

# On Study of Two-Dimensional Lunar Scan for Advanced Technology Microwave Sounder Geometric Calibration

Jun Zhou[1] and Hu Yang[1]

[1]Earth System Science Interdisciplinary Center (ESSIC), University of Maryland, College Park, MD, 20740, USA

**Correspondence:** Hu Yang (huyang@umd.edu)

**Abstract.** The NOAA-20 satellite was successfully launched on 18 November 2017. It carries five key instruments including Advanced Technology Microwave Sounder (ATMS). On January 31, 2018, the spacecraft performed a pitch-over maneuver operation, during which the two-dimensional lunar scan observations were collected. In this study, a technique has been developed by which the ATMS on-orbit geometric calibration accuracy can be validated based on this lunar scan dataset. The fully

calibrated data are fitted in the antenna pattern coordinate by Gaussian function. The deviation of the center of the fit function from the origin of the frame is taken to be the boresight pointing error of the instrument. This deviation is further transformed to the Euler angle roll and pitch defined in spacecraft coordinate system. The estimated ATMS boresight pointing Euler angle roll (pitch) is 0.05° (0.22°) at K-band, -0.07° (0.25°) at Ka-band, 0.02° (0.24°) at V-band, -0.07° (-0.08°) at W-band, and -0.04° (0.02°) at G-band. The results are validated by comparing with those derived from the coastline inflection point method.

It shows that the Euler angles derived from these two independent methods are consistent very well. For the sounding channels where the coastline method is inapplicable, the lunar scan method is still capable of delivering reasonable estimations of their geometric calibration errors.

## 1 Introduction

On 18 November 2017, NOAA-20 satellite was launched to a polar orbit at 824 km above the Earth with an inclination angle

of 98.7°, leading its predecessor Suomi NPP (SNPP) by half an orbit. It inherits all the five key instruments from SNPP, that is, the Advanced Technology Microwave Sounder (ATMS), the Cross-track Infrared Sounder (CrIS), the Ozone Mapping and Profiler Suite, the Visible Infrared Imaging Radiometer Suite (VIIRS), and the Clouds and the Earth's Radiant Energy System (CERES). ATMS is a cross-track scanning microwave radiometer, providing total of 22 channels at microwave frequency ranging from 23 to 183 GHz for profiling the atmospheric temperature and moisture under all weather conditions. As part

of the ATMS calibration and validation activities, the geolocation accuracy of ATMS data must be well characterized and documented during post-launch.

  Most of the methods applied to assess the on-orbit geolocation error of microwave sensors rely on Earth targets, such as coastline inflection point method (CIP) (Hoffman et al., 1987; Smith et al., 2009; Gregorich and Aumann, 2003; Currey, 2002), image co-registration method (Wang et al., 2013, 2017; Wolfe et al., 2002, 2013; Khlopenkov et al., 2010), land-sea fraction

method (LFM) (Bennartz, 1999), and ascending and descending observation comparison (Moradi et al., 2013). Recent study





(Zhou et al., 2019) disclosed that antenna beam misalignment is a major error source in ATMS total geolocation error budget. This static error term causes ATMS boresight pointing error to have a scan-angle dependent feature that should be corrected by Euler angle roll and pitch determined at each field-of-view (FOV) position. In that study, the coastline inflection point method has been improved by taking this into account. In addition, the coastlines along in-track and cross-track direction are carefully

selected to assess the geolocation error in a more accurate way. By taking these measures, the retrieval error can be reduced below 10% and the scan-angle dependent feature of geolocation error is also largely mitigated. Based on the retrieved Euler angles, correction matric can be built and applied into the operational geolocation process to enhance the on-orbit geolocation accuracy.

However, the application of these earth-target dependent methods has some limitations. Firstly, they can only be applied

to the window channels of microwave radiometers as sounding channels cannot see Earth surface due to strong atmospheric absorption. Secondly, the existence of clouds will shade the Earth target, and thus the cloud contaminated pixels need to be screened out. Regarding the aforementioned issues, it is very necessary to develop an on-orbit boresight pointing accuracy evaluation algorithm which is completely independent of the Earth targets.

As a distinctive target with stable microwave emission in the cold cosmic background, the Moon has already been proved

to be very useful in evaluating ATMS long-term calibration stability. In doing so, a physical model is developed to simulate the lunar emission at microwave frequencies (Yang et al., 2018). However, the possibility of using the Moon for geolocation validation and correction of microwave sensors has not been widely discussed. Attempt has been made to utilize the lunar intrusion data to assess the boresight pointing error of Microwave Humidity Sounder (MHS) (Burgdorf et al., 2016) and ATMS (Zhou et al., 2017). But as the Moon only passes the along-track direction of the four space view positions during lunar

intrusion, the boresight pointing information the data can provide is very limited.

On January 31, 2018, spacecraft pitch maneuver operation was carried out for NOAA-20 satellite, when lunar disk is in a full moon phase. The spacecraft was pitched completely 360 ° in about 14 minutes when satellite entered the Earth umbra region, thereby enabling all the instruments to acquire full scans of deep space. For ATMS, this maneuver establishes a baseline radiometer output from pure cold space. In the middle of pitch-over operation, a full-Moon disk radiation flux was captured at

pitch angle around 179° and data was collected for all channels of ATMS. This two-dimensional lunar scan observation dataset provides a unique chance to study the ATMS radiometric and geometric calibration accuracy.

The objective of this study is to develop an algorithm to assess ATMS boresight pointing accuracy based on the two-dimensional lunar scan observations during pitch-over maneuver operation. This paper is organized as follows: Sect. 2 briefs ATMS instrument scan geometry and geolocation process, Sect. 3 presents the lunar scan observations captured by ATMS

during the pitch-over maneuver operation, Sect. 4 describes the methodology, Sect. 5 presents the results and the validation with the coastline method developed in (Zhou et al., 2019), conclusions and discussions are in Sect. 6.



## 2 ATMS geolocation algorithm

### 2.1 Scan geometry

During the process of pitch maneuver, the ATMS instrument performed a normal scan and data from the Earth view, deep space view(DSV) and internal calibration target (ICT) view were collected. For each ATMS scan cycle, the earth is viewed at

96 different scan angles, which are distributed symmetrically around the nadir direction. Such 96 ATMS field-of-view (FOV) samples are taken with each FOV sample representing the mid-point of a brief sampling interval of about 18 ms (JPSS ATMS SDR Calibration ATBD, 2013). With a scan rate of 61.6° per second, the angular sampling interval is 1.11°. Therefore, the angular range between the first and last (i.e., 96th) sample centroids is 105.45° (i.e., 52.725° relative to nadir). At each scan, while there are four warm load samples being taken at angle of 193.3°, 194.4°, 195.5° and 196.6°, four deep space view

samples can also be collected around 83.4°, with each sample spaced 1.11 degrees apart (Yang and Weng, 2016). The Moon's disk was captured by ATMS at all channels during the pitch-over maneuver operation. The Moon appears between ATMS FOV 60 to 70 in about ±20 scan lines around center peak pitch angle of 180°. With the DSV and ICT samples at each scan, the collected raw data counts of lunar scans were being able to transferred to lunar radiation flux by using the two-point calibration equation (Yang et al., 2016).

### 2.2 Geolocation algorithm

The goal of the ATMS geolocation algorithm is to map the beam pointing vector to geodetic longitude and latitude on the Earth ellipsoid for each FOV at each scan position. Specifically, the ATMS geolocation process includes an instrument geolocation module and a common geolocation module. In the instrument geolocation module, the sensor exit vector in the antenna coordinate system is built from scan angle $\vartheta$:

$$\boldsymbol{b_{Ant}} = \begin{pmatrix} 0 \\ \sin\vartheta \\ \cos\vartheta \end{pmatrix} \qquad (1)$$

$\boldsymbol{b_{Ant}}$ is then transformed to the spacecraft coordinate system (SC) by applying the antenna beam misalignment correction matrix $ROT_{Inst/Ant}$ and the instrument mounting matrix $ROT_{SC/Inst}$. The antenna beam alignment with respect to the instrument cube was measured as Euler angles during the antenna subsystem verification test for each channel at three different scan positions (FOV-1, 48, and 96) and interpolated to the other positions. Instrument mounting error is defined as the misalign-

ment between the instrument and the spacecraft coordinate systems. It is measured in terms of Euler angles during prelaunch ground test (JPSS ATMS Calibration Data Book, 2007). The corrected vector continues to go through the common geolocation module which is shared among all the sensors onboard NOAA-20. In this module, the beam vector is first transformed from the spacecraft coordinate system to Earth-centered initial (ECI) coordinate system through the matrix $ROT_{ECI/SC}$ built from the quaternions of spacecraft attitude. The rotation matrix $ROT_{ECEF/ECI}$ is then applied to transform the beam vector from ECI

to the Earth-centered Earth-fixed (ECEF) coordinate system by taking Earth orientation, including polar wander, procession,





and nutation, into account. Finally, the geodetic latitude and longitude for each FOV can be derived from the intersection of the corrected sensor exit vector $b_{ECEF}$ with WGS84 reference frame (Baker, 2011). The complete geolocation algorithm is showed in the following equation:

$$b_{ECEF} = ROT_{ECEF/ECI} ROT_{ECI/SC} ROT_{SC/Inst} ROT_{Inst/Ant} b_{Ant} \qquad (2)$$

5 The antenna beam pointing direction is differentiated into five bands: K- (channel 1), Ka- (channel 2), V- (channels 3–15), W- (channel 16), and G- (channels 17–22) bands. Each band has its own set of latitude and longitude at each beam position.

This study focuses on the evaluation and correction of the static error mainly originating from the antenna beam misalignment and the instrument mounting error which are the dominant part in the total geolocation error budget. Even though these static error terms have been measured in the prelaunch ground test and the corrections have been included in geolocation 10 process, residual errors may still exist due to the on-orbit thermal dynamic change and shift during and after the launch.

When operating in normal mode, the spacecraft coordinate system is fixed to the spacecraft with its origin at the spacecraft center of mass, Z-axis ($Z_{SC}$) pointing towards nadir, X-axis ($X_{SC}$) pointing to the along-track direction, and Y-axis ($Y_{SC}$) completing the right-hand coordinate system. While during the pitch-over maneuver operation, the satellite was rotating around $Y_{SC}$, as is showed in Fig. 1. For ATMS, the scan is performed in y-z plane of the antenna coordinate system. As the antenna 15 and instrument coordinates are aligned with the spacecraft frame, the pitch-over maneuver operation made the scan plane rotate around $Y_{SC}$ with the satellite. When the scan plane rotated towards the Moon, a full lunar disk radiation flux was captured and the data was collected for all channels of ATMS. The observed raw data counts were transferred to lunar radiation flux by

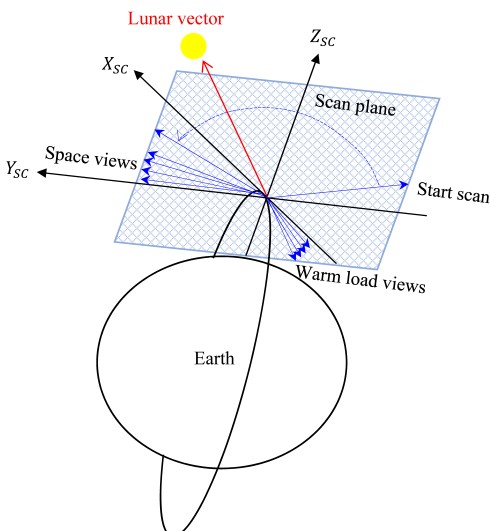

**Figure 1.** ATMS scan geometry when operating in pitch-over mode. $X_{SC}$, $Y_{SC}$, and $Z_{SC}$ are the axes of spacecraft coordinate system and its Y-Z plane is the scan plane. The lunar vector entered the scan plane as the spacecraft rotated about 180° pitch angle around $Y_{SC}$.

using the calibration equation with the warm load brightness temperature and cold space brightness temperature being further





corrected for warm bias, earth side lobe contamination, as well as the reflector emission contamination (Yang et al., 2016). To derive the pure lunar signal, the cosmic background radiation is subtracted from the calibrated brightness temperature. Fig. 2 shows the observed lunar brightness temperature for channel 1, 3 and 17. Note that in Fig. 2, since the beam width is different for different channel, the lunar disk extended over more scan lines and FOVs for lower frequency channels (channel 1) than higher frequency channels (Channel 3 with 2.2° and channel 17 with 1.1°).

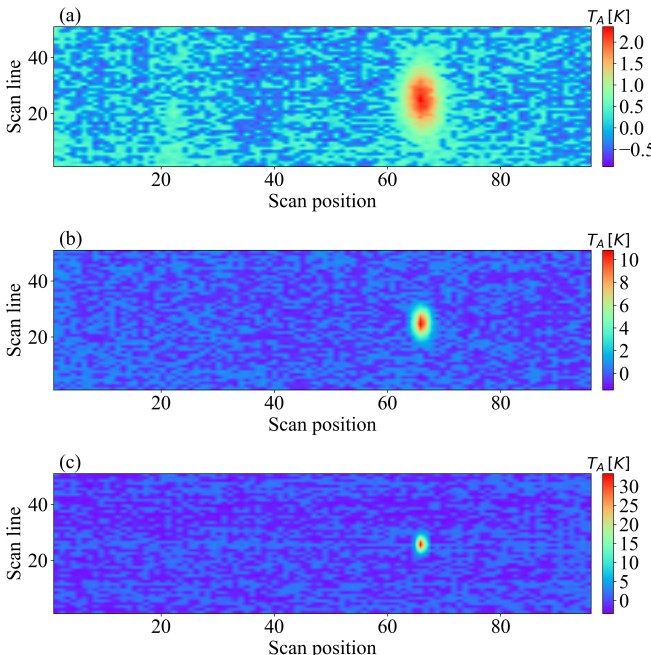

**Figure 2.** Calibrated lunar antenna temperature $T_A$ for channel 1(a), 3(b), and 17(c).

## 3 Methodology

The Lunar scan data can be expressed as the integration of antenna response weighted Moon radiance over solid angle $\Omega_{moon}$ and sampling time $\tau$:

$$T_A = \frac{1}{\tau \cdot \Omega_A} \int_{-\frac{\tau}{2}}^{\frac{\tau}{2}} \iint_{\Omega_{moon}} Tb_{moon}^{disk} \cdot G_{ant}(\theta,\varphi) \cdot \sin\theta d\theta d\varphi \qquad (3)$$

10   where $\Omega_A$ is a beam solid angle, $Tb_{moon}^{disk}$ is the brightness temperature of the Moon disk, and $G_{ant}(\theta,\varphi)$ is ATMS antenna pattern. $Tb_{moon}^{disk}$ depends on the effective surface emissivity and the phase angle of the Moon. As the change of the phase angle during the pitch-over operation is less than 0.001°, $Tb_{moon}^{disk}$ can be regarded as a constant for a certain channel and taken out of





the integration. Therefore, the lunar observations can be modeled as the scaled integration of antenna response. For ATMS, the antenna response has a Gaussian-like distribution with its peak locating at the center of the antenna pattern frame. Accordingly, the lunar scan data should appear the same shape in that frame except that the spread of the Gaussian function is widened by the integration over solid angle and sampling time. More details about the lunar modeling can be found in (Yang et al., 2018).

5      In this section, the method to project lunar observations into the antenna pattern frame is presented, followed by the development of the algorithm to derive ATMS boresight pointing error from the projected lunar scan data.

### 3.1   Determination of lunar position in antenna pattern coordinates

Antenna pattern coordinate system, where the antenna response is measured in the pre-launch ground test, is defined with its X-axis ($X_{AntPattn}$) pointing to the along-track direction, Z-axis ($Z_{AntPattn}$) being aligned with the beam vector, and Y-axis
10   ($Y_{AntPattn}$) completing the right-hand coordinate system. The zenith and azimuthal angles of the Moon vector in this frame, showed as the $(\theta, \varphi)$ in black in Fig. 3, can be determined through the following process.

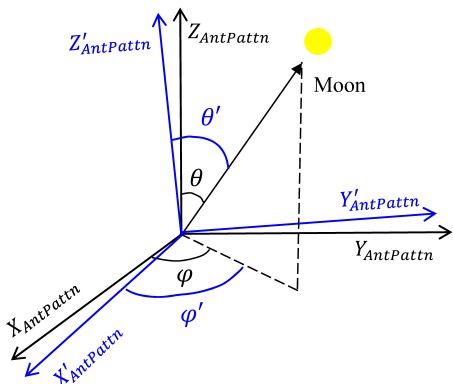

**Figure 3.** Schematic illustration of the Moon vector in antenna pattern frame. $(X'_{AntPattn}, Y'_{AntPattn}, Z'_{AntPattn})$ are the real axes of the frame and $(X_{AntPattn}, Y_{AntPattn}, Z_{AntPattn})$ are the nominal axes with beam boresight pointing error. $(\theta', \varphi')$ and $(\theta, \varphi)$ are the zenith and azimuthal angles of the Moon vector determined in the real and nominal frames respectively.

Given the ephemeris of the Moon, the Lunar vector in ECI ($\boldsymbol{l_{ECI}}$) at any observation time can be calculated through the Naval Observatory Vector Astrometry Software Package version F3.1 (NOVAS F3.1). This software package is an open source library for computing various commonly used quantities in positional astronomy (Kaplan et al., 2011). $\boldsymbol{l_{ECI}}$ is then
15   transformed back to antenna coordinate system by applying the transpose of the rotation matrices in operational geolocation process Eq. (2):

$$\boldsymbol{l_{Ant}} = ROT^T_{Inst/Ant} ROT^T_{SC/Inst} ROT^T_{ECI/SC} \boldsymbol{l_{ECI}} \tag{4}$$

To further transform $\boldsymbol{l_{Ant}}$ into antenna pattern coordinate system, the rotation matrix $ROT_{AntPattn/Ant}$ needs to be created. This can be done by defining the three axes of antenna pattern coordinate in antenna coordinate system. According to its





definition, $X_{AntPattn}$ is aligned with $X_{Ant}$, $Z_{AntPattn}$ is aligned with $\boldsymbol{b_{Ant}}$, and $Y_{AntPattn}$ is the cross product of $Z_{AntPattn}$ and $X_{AntPattn}$. Therefore, the rotation matrix $ROT_{AntPattn/Ant}$ can be built as below:

$$\boldsymbol{X_{AntPattn}} = \begin{pmatrix} 1 \\ 0 \\ 0 \end{pmatrix} \tag{5}$$

$$\boldsymbol{Z_{AntPattn}} = \boldsymbol{b_{Ant}} = \begin{pmatrix} 0 \\ \sin\vartheta \\ \cos\vartheta \end{pmatrix} \tag{6}$$

$$\boldsymbol{Y_{AntPattn}} = \boldsymbol{Z_{AntPattn}} \times \boldsymbol{X_{AntPattn}} \tag{7}$$

$$ROT_{AntPattn/Ant} = \begin{pmatrix} \boldsymbol{X_{AntPattn}} & \boldsymbol{Y_{AntPattn}} & \boldsymbol{Z_{AntPattn}} \end{pmatrix}^T \tag{8}$$

Then the lunar vector in antenna pattern coordinate system can be derived:

$$\boldsymbol{l_{AntPattn}} = ROT_{AntPattn/Ant}\boldsymbol{l_{Ant}} \tag{9}$$

The zenith and azimuthal angle of the lunar vector in polar coordinate system of antenna pattern can be obtained:

$$\theta = \arctan\left( \frac{\sqrt{\boldsymbol{l_{AntPattn}}[1]^2 + \boldsymbol{l_{AntPattn}}[2]^2}}{\boldsymbol{l_{AntPattn}}[3]} \right) \tag{10}$$

$$\varphi = \arctan\left( \frac{\boldsymbol{l_{AntPattn}}[2]}{\boldsymbol{l_{AntPattn}}[1]} \right)$$

To present the Gaussian-like distribution of observations, the data are projected in Cartesian coordinate:

$$x = \sin\theta\cos\varphi$$

$$y = \sin\theta\sin\varphi \tag{11}$$

The observations of channel 1, 3, and 17 projected in antenna pattern coordinate are plotted on Fig. 4 (a), (c), and (e). To extract the valid lunar signal, the observations with the zenith angle larger than a certain value where the negative observations appear are excluded from the dataset. Because of the difference in beam width, the FOV positions that can detect the Moon radiance range from FOV 63 - 70 for channel 1-2, FOV 65 - 68 for channel 3-16, and FOV 65 - 67 for channel 17 -22. The FOV 66 is the scan position where the center of the Moon appears closest to the center of FOV.

## 3.2 Development of the evaluation algorithm

As described in Sect. 3.1, the key issue in determining the position of the Moon in antenna pattern is to correctly establish the antenna pattern frame based on the pointing direction of beam vector. If the boresight pointing error exists, as the case showed in Fig. 3, the beam misalignment will make the actual antenna pattern coordinate $(X'_{AntPattn}, Y'_{AntPattn}, Z'_{AntPattn})$ deviate from its nominal position $(X_{AntPattn}, Y_{AntPattn}, Z_{AntPattn})$. The deviation of antenna pattern frame leads to errors in the



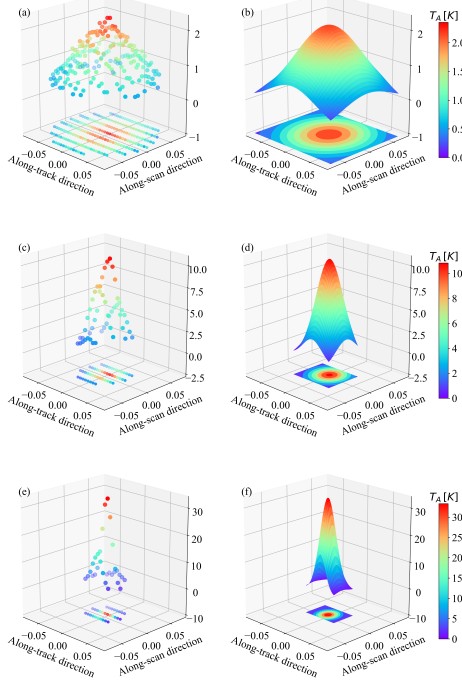

**Figure 4.** Raw observations (the first column) and their fitted two-dimensional Gaussian function (the second column) of channel 1 (a-b), 3 (c-d), and 17 (e-f) in antenna pattern coordinate system.

determined zenith and azimuthal angles of the Moon vector. Through Eq. (4-11), the Lunar vector is projected into the nominal antenna pattern frame, not the actual one where the observations are carried out. Consequently, the maxima of the observations in the nominal antenna pattern frame will shift from the coordinate origin.

The basic idea of using Lunar scan data to identify the ATMS geometric calibration error is to correct the nominal antenna pattern frame until it is perfectly aligned with the actual one, under which condition the maxima of the observations would be locating at the origin of the antenna pattern frame. The correction can be done by correcting each axis of the antenna pattern frame with the Euler angles counteracting the geometric calibration error. To maintain the consistency with the coastline method (Zhou et al., 2019), in which the boresight pointing error is defined in terms of Euler angle roll and pitch at each FOV in spacecraft coordinate system, the axes of antenna pattern coordinate system need to be transformed from antenna coordinate system to spacecraft coordinate system by going through the rotation matrices in Eq. (2), corrected, and then transformed back to the antenna coordinate system:

$$X'_{AntPattn} = ROT^T_{Inst/Ant} ROT^T_{SC/Inst} ROT_{corr} ROT_{SC/Inst} ROT_{Inst/Ant} X_{AntPattn} \qquad (12)$$

where $ROT_{corr}$ is the correction matrix. Since the FOV 66 is the scan position where the lunar vector is closest to the FOV center during the pitch-over maneuver operation, the beam pointing error of that specific FOV is supposed to have the strongest contribution to the shift of the observation peak in antenna pattern frame. Therefore, the elements of $ROT_{corr}$ are defined by





Euler angle roll $\xi_r$ and pitch $\xi_p$ in spacecraft coordinate system at the 66th scan position:

$$ROT_{corr} = ROT_r(\xi_r) \cdot ROT_p(\xi_p)$$

$$ROT_r(\xi_r) = \begin{pmatrix} 1 & 0 & 0 \\ 0 & \cos\xi_r & -\sin\xi_r \\ 0 & \sin\xi_r & \cos\xi_r \end{pmatrix}$$

$$ROT_p(\xi_p) = \begin{pmatrix} \cos\xi_p & 0 & \sin\xi_p \\ 0 & 1 & 0 \\ -\sin\xi_p & 0 & \cos\xi_p \end{pmatrix} \tag{13}$$

The same correction process is applied to $Z_{AntPattn}$. After the vectors of the x- and z-axis of antenna pattern coordinate system in antenna frame are corrected, the rotation matrix can be updated through Eq. (7-8) and the lunar vector in antenna pattern frame can be relocated through Eq. (9-11). Since the observations are made at discrete points, the Moon may not pass the center of the antenna pattern during the pitch-over maneuver operation. To determine the position of the lunar vector in the antenna pattern frame where the Moon reaches the center of antenna pattern, the observations are fitted by the two-dimensional

Gaussian function:

$$f(x,y) = A \cdot \exp\left(-\left(\frac{(x-x_0)^2}{2\sigma_x^2} + \frac{(y-y_0)^2}{2\sigma_y^2}\right)\right) \tag{14}$$

where A is the amplitude, $(\sigma_x, \sigma_y)$ are the Gaussian RMS width along x-, and y-axis direction, and $(x_0, y_0)$ is the position of the center of the Gaussian function. The cost function can be defined as:

$$\varepsilon = \sqrt{x_0^2 + y_0^2} \tag{15}$$

The fitting can also help to reduce the observation noise, especially for channel 1 and 2 whose signal-to-noise ratio are higher than other channels. The fitting results are presented in Fig. 4 (b), (d), and (f).

    The steps of the evaluation algorithm are summarized as below. Given an initial value of Euler angles $(\xi_r, \xi_p)$, the axes of the nominal antenna pattern frame (Eq. (5-6)) are corrected through Eq. (12-13). Then the rotation matrix $ROT_{AntPattn/Ant}$ defined in Eq. (8) is updated, through which the Moon in antenna pattern frame is relocated by applying Eq. (9-11). The

observations projected in antenna pattern frame are fitted by two-dimensional Gaussian function (Eq. (14)) and the center of the function is used to calculate the cost function (Eq. (15)). The Euler angles are adjusted and the above process is repeated until the cost function reaches its minima. The Euler angles that corresponding to the minima of the cost function are the estimated boresight pointing error.

## 4   Results and validation

The evaluation algorithm developed in Sect. 3 is applied to ATMS lunar scan observations obtained during the NOAA-20 pitch-over maneuver operation. Fig. 5 shows the variation of the cost function with the Euler angle roll and pitch being tuned





from -1° to 1° with 0.01° interval. For each channel, a unique minimum value of the cost function is found and the pair of roll and pitch angle at that point is taken to be the boresight pointing error for that channel.

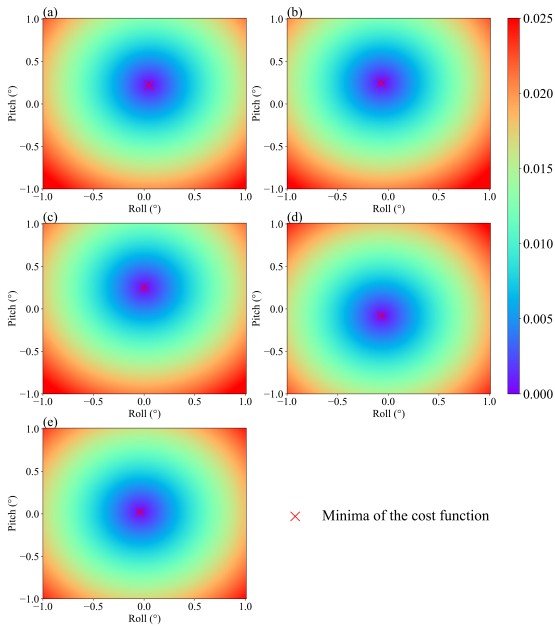

**Figure 5.** Variation of cost function with respect to the shifting of the Euler angle roll and pitch for ATMS channel 1 (a), 2 (b), 3 (c), 16 (d), and 17 (e). The red crosses represent the points where the minima of the cost functions are found.

The estimated boresight pointing error for ATMS total 22 channels are presented in Fig. 6. It is noticeable that channel 1 - 15 have large boresight pointing error in pitch direction that is up to 0.25° while the error in roll direction of these channels and the error in roll and pitch directions of other channels are below 0.08°. The retrieved Euler angles of the channels in each band are quite stable, which coincides with the fact that the channels in each band share the same geolocation position. The Euler angle roll (pitch) averaged over the channels of each band is 0.05° (0.22°) at K-band, -0.07° (0.25°) at Ka-band, 0.02° (0.24°) at V-band, -0.07° (-0.08°) at W-band, and -0.04° (0.02°) at G-band.

To independently validate the boresight pointing error estimated from the Lunar scan observations, the retrieved Euler angle roll and pitch at FOV 66 are compared with those retrieved from the coastline inflection point method (Zhou et al., 2019). The results are presented in Fig. 7. Note that in coastline method, the Euler angle roll and pitch at each FOV position is retrieved from the samples around that specific position. To reduce the uncertainty, the retrieved roll and pitch of each channel is then fitted by a quadratic polynomial function and the Euler angle at each FOV on the fitting line is taken as the final solution, which is showed as the black and blue lines on the panels of Fig. 7. The Euler angle roll and pitch at FOV 66 derived from the lunar observations are plotted as the red and magenta crosses on the same panel. It can be seen that the Euler angles retrieved from these two independent methods are consistent with each other very well. The difference between them are within the





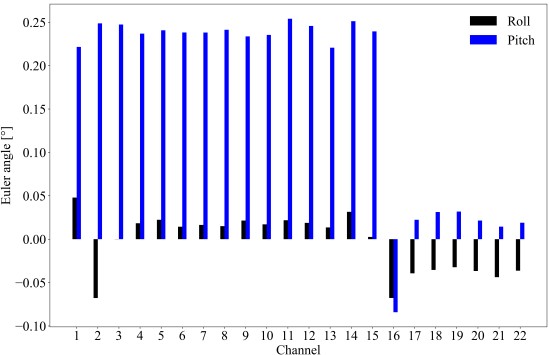

**Figure 6.** Retrieved Euler angle roll (black bar), and pitch (blue bar) for ATMS channel 1-22.

uncertainty of the retrieval results of coastline method. For the sounding channels where the coastline inflection point method is inapplicable, the lunar scan method is still capable of delivering reasonable estimations of the geometric calibration errors.

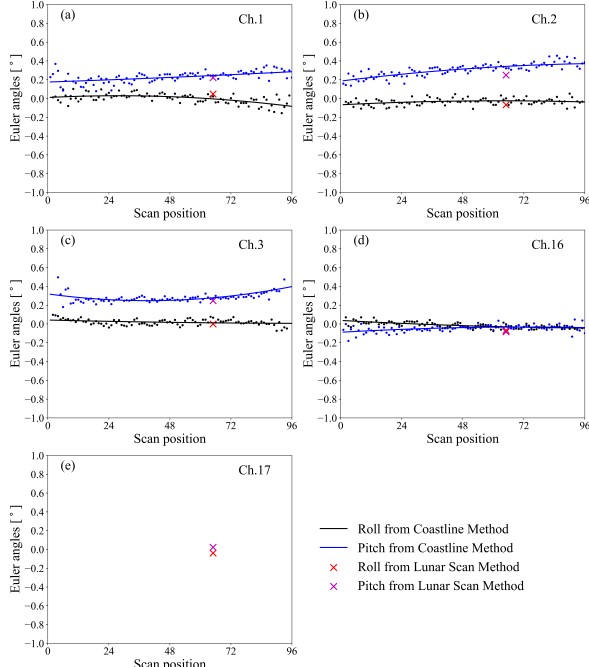

**Figure 7.** Comparison between the Euler angles retrieved from coastline inflection point method and those from lunar scan method.





## 5   Conclusions and discussions

In this study, a unique two-dimensional lunar scan dataset from Advanced Technology Microwave Sounder (ATMS) onboard NOAA-20 was studied and an algorithm was developed for validating ATMS pointing accuracy based on this dataset. Beam pointing errors were derived in terms of Euler angle roll and pitch defined in spacecraft coordinate system, so that they can be

directly compared with those retrieved by the coastline method proposed in our previous study. Existing research showed that ATMS boresight pointing error has scan-angle dependent feature due to the dominant role played by antenna beam misalignment in total error budget. As the Moon signal has the most significant impact on the 66th FOV during the pitch-over maneuver, the boresight pointing error assessed by this lunar scan dataset is regarded as the error at that specific scan position. Retrieving results show that NOAA-20 ATMS beam misalignment in terms of Euler angle roll (pitch) at the 66th FOV is estimated to be

0.05° (0.22°) at K-band, -0.07° (0.25°) at Ka-band, 0.02° (0.24°) at V-band, -0.07° (-0.08°) at W-band, and -0.04° (0.02°) at G-band. These results are consistent with those derived by coastline method. For the sounding channels where the coastline method is inapplicable, the lunar scan method is still capable of delivering reasonable estimations of the geometric calibration errors.

The advantage of this method is that it can be used as a unique method to assess the instrument beam pointing error during

the post-launch instrument early check-up phase. It can also serve as a supplement for the earth-target based technique to evaluate the geometric calibration accuracy of sounding channels. For window channels, it can provide a cross-check of the methods relying on ground reference.

For NOAA-20 operating in an afternoon orbit, the pitch-maneuver brings the Moon into ATMS scan plane at a specific scan angle and thus only the beam pointing error at that specific FOV position can be derived through the algorithm developed in

this study. If the roll-over maneuver is carefully designed and carried out at appropriate times, allowing the Moon to enter the scan plane at different scan angles, the pointing error at other FOVs can be evaluated as well.

*Data availability.*   The NOAA-20 ATMS data during pitch-over maneuver are available from NOAA CLASS,
$https://www.bou.class.noaa.gov/saa/products/welcome; jsessionid = D424F0127A4B97AD2AFF957AA2B15687$. Some of them are displayed on STAR ICVS, $https://www.star.nesdis.noaa.gov/icvs/index.php$. The Lunar vector is calculated through the

Naval Observatory Vector Astrometry Software (NOVAS) Package, which is a open source library provided by
$https://aa.usno.navy.mil/software/novas/novas_info.php$. The instrument beam vector and rotation matrices of ATMS geolocation process are output from Algorithm Development Library (ADL) software, $https://jpss.ssec.wisc.edu/$.

*Author contributions.*   H. Yang started and guided this investigation. J. Zhou developed the retrieval algorithm. H. Yang and J. Zhou prepared the manuscript.



*Competing interests.* The authors declare that they have no conflict of interest.



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



Joint Polar Satellite System (JPSS) Advanced Technology Microwave Sounder (ATMS) SDR Calibration Algorithm Theoretical Algorithm Theoretical Basis Document (ATBD), Center for Satellite Applications and Research, College Park, Maryland, December 18, 2013.

Joint Polar Satellite System (JPSS) Advanced Technology Microwave Sounder (ATMS) Calibration Data Book, document ATMS PFM P/N 1362460-1 S/N 302, Northrop Grumman, Azusa, CA, USA, Mar. 2007.