# Peer review of "On Study of Two-Dimensional Lunar Scan for Advanced Technology Microwave Sounder Geometric Calibration"

_Atmospheric Measurement Techniques, 2019_

## Short Comment (SC1) · 17 May 2019

The satellite position is not taken into account for the Moon viewing direction computation, i.e. eq. 4 transforms the Moon position from ECI (Earth-Centred Inertial) to antenna frame (satellite centred) without keeping into account the shift between the two frames and that this shift changes during the scans. But likely, considering the distance to the Moon, the effect is negligible.
* * *

---

## Author Comment (AC1) · 20 May 2019

This is a very good question. As showed in Figure 1 of the manuscript, the lunar vector in this study is always pointing from the satellite center to the Moon center. By setting appropriate input parameters, this vector in ECI can be calculated through the function PLACE in NOVAS package. Please refer to Kaplan et al., 2011, page 40-43 for details. The I_ECI appeared in Equation 4 is the lunar vector originating from satellite center, not from Earth center. The notation may lead to some ambiguity. We will clarify that in the revised version of the manuscript. Thank you very much for pointing out this.

---

## Referee Comment (RC1) · Anonymous Referee #1 · 3 Jun 2019

General Comments

The manuscript demonstrates how a raster scan of the Moon during a pitch-over manoeuvre can be used for calculating and correcting the boresight pointing error of a microwave sounder. This method is new and relevant for the check-out phase of future microwave sounders, because it makes a good case for pitch-over manoeuvres. It was correctly applied by the authors and their approach is described in a convincing way.

Specific Comments

I see two potential issues with the method the authors employ:

[Figure]

1. De Bartolomei's comment is not completely addressed by Zhou's answer. The Moon describes a full circle in the sky in about one month, i. e. it moved by about 0.1 degrees during the 14 minutes of the pitch manoeuvre. This is more than most of the misalignments the authors found. How long did it take to carry out the two-dimensional lunar scan observation? How can the authors be sure they calculated the variation of the cost function correctly, when the Moon moved during the observation?

2. The authors give no explanation for the surprising fact that the boresight pointing error of channels 1 - 15 in pitch direction is three times as high as all other errors. Could it be that this error is an artefact caused by the time constant of the receivers V and K/Ka? If the devices react with a little delay to changes in the incoming flux, then they will give their maximum signal only after the Moon was at the origin of the antenna pattern frame.

Technical Corrections:

Page 1, line 15: The authors only explain what the "S" in "SNPP" stands for - it is better to explain the whole abbreviation.

Page 2, lines 17 and 19: I suggest to cite also R. Bonsignori, In-orbit verification of microwave humidity sounder spectral channels coregistration using the moon, Applied Remote Sensing, Volume 12, id. 025013 (2018).

Page 2, lines 19 and 20: I do not understand "the Moon only passes the along-track direction".

Page 2, line 31: There is no Sect. 6.

Page 7, line 16: What are "negative observations"?

Figure 4: Units of axes?

Page 9, line 15: "lower" instead of "higher"?

Figure 5: The red X is hard to see.
Figure 7: The colours of the two Xs are almost the same.

[Figure]

---

## Referee Comment (RC2) · Anonymous Referee #2 · 5 Jun 2019

General Comments

This paper gives a good overview of the application of using the moon as a calibration reference for determining geolocation, which is necessary for microwave radiometers and as noted by the authors is difficult for channels that do not see Earth's surface. A detailed explanation of the method is given and results shown applying the method to ATMS. In addition to the comments already noted by the other reviewers, I have the following comments below.

Specific Comments

The Figure 7 coastline Euler angles do not match those shown in Zhou et al. 2019

[Figure]

Figure 9. Can you please provide an explanation for this? Since Figure 7 provides validation for your lunar method and makes a direct reference to the Zhou et al. 2019 paper, I'm assuming Figure 9 in that paper is what you are referring to and these figures should then match unless you explain the discrepancies.

Another comment in regards to Figure 7, is that it would be good to show a table listing the coastline Euler angles from your previous paper at FOV 66, compared with these numbers along with the difference to quantitatively show how the methods compare rather than just the figure.

Technical Corrections

Page 2, line 7. Change "matric" to "metric".

Page 4, line 3. Change "showed" to "shown".

Page 6, line 2. Change "locating" to "located".

Page 6, line 11. Change "showed" to "shown".

Page 8, line 6. Change "locating" to "located".

Page 9, line 22. Change "corresponding" to "correspond".

---

## Author Comment (AC2) · 18 Jun 2019

Dear Referee 1 and Editors,

Thank you very much for your valuable comments and suggestions. The reply to them along with the revised manuscript have been combined to a single pdf and uploaded as a Supplement. Please do not hesitate to contact me (jzhou128@umd.edu) if you have any questions.

Best regards,

Jun Zhou and Hu Yang 6/18/2019

[Figure]

Please also note the supplement to this comment:
https://www.atmos-meas-tech-discuss.net/amt-2019-177/amt-2019-177-AC2-supplement.pdf
* * *
[Figure]

**Supplement:**

General Comments

The manuscript demonstrates how a raster scan of the Moon during a pitch-over maneuver can be used for calculating and correcting the boresight pointing error of a microwave sounder. This method is new and relevant for the check-out phase of future microwave sounders, because it makes a good case for pitch-over maneuvers. It was correctly applied by the authors and their approach is described in a convincing way.

Thank you very much for your valuable comments and suggestions. They are very helpful for us to improve the lunar scan algorithm and the quality of the manuscript. We've studied your comments thoroughly and been able to respond to each of them as below. The manuscript has been revised accordingly and the changes made to it have been marked by using latexdiff.

Specific Comments

I see two potential issues with the method the authors employ:

1. De Bartolomei's comment is not completely addressed by Zhou's answer. The Moon describes a full circle in the sky in about one month, i. e. it moved by about 0.1 degrees during the 14 minutes of the pitch maneuver. This is more than most of the misalignments the authors found. How long did it take to carry out the two-dimensional lunar scan observation? How can the authors be sure they calculated the variation of the cost function correctly, when the Moon moved during the observation?

Reply: Thank you for your question. During the pitch maneuver operation, about $\pm 20$ scan lines are affected by the Moon and that is equivalent to about 2 minutes. It is true that both the Moon and the satellite moved during this 2-minute period. The algorithm developed in this paper has already taken that into consideration.

As stated in Section 3, the key to calculate the cost function is to determine lunar position vector pointing from satellite to the Moon in antenna pattern coordinate system. The algorithm begins with calculating the lunar vector in ECI through the Naval Observatory Vector Astrometry Software Package version F3.1 (NOVAS F3.1). This software package is an open source library for computing various commonly used

quantities in positional astronomy (Kaplan et al., 2011). It has been implemented in the Algorithm Development Library (ADL), a software program for the NOAA polar-orbiting satellite data process, to calculate the Sun and the Moon vectors in order to determine the zenith and azimuthal angles of the Sun and set the Lunar intrusion flag (https://jpss.ssec.wisc.edu/). The process is summarized as below.

For a specific observation time, the Earth position vector $\overrightarrow{PEB}$ and the Moon position vector $\overrightarrow{POS_1}$ referred to the International Celestial Reference System (ICRS) are interpolated from their ephemeris provided by file "SS_EPHEM.TXT". ICRS is the barycentric coordinate system with its x-axis pointing to the dynamical equinox of J2000, z-axis being aligned with the Earth spin axis, and y-axis completing the right-hand coordinate system. The satellite position vector $\overrightarrow{POG}$ in ECEF can be interpolated from GPS observations and then transformed into Geocentric Celestial Reference System (GCRS), i.e. ECI. GCRS is a geocentric coordinate system oriented according to the ICRS axes. Therefore, the transformation between ICRS and GCRS coordinates contains no rotation components (IERS Technical Note No. 36. https://www.iers.org/SharedDocs/Publikationen/EN/IERS/Publications/tn/TechnNote36/tn36_174.pdf?__blob=publicationFile&v=1). The satellite position vector is then transformed from GCRS to ICRS:

$$\overrightarrow{POB} = \overrightarrow{PEB} + \overrightarrow{POG}$$

The Lunar vector originating from the satellite to the Moon in GCRS can be calculated:

$$\overrightarrow{l_{ECI}} = \overrightarrow{POS_1} - \overrightarrow{POB}$$

The light-time correction, gravitational light bending, and aberration are also considered in the process.

[Figure]

Fig.1 The geometry in the calculation of the lunar vector $\overrightarrow{l_{ECI}}$ by NOVAS package.

For the details, please refer to the subroutine PLACE in NOVAS package and ProSdrCmnGeo.cpp in ADL 5.3.

The lunar vector is then transformed from ECI to antenna pattern coordinate system ($l_{AntPattn}$) through equation (4)-(9) in the manuscript. The zenith and azimuthal angles of the Moon in the antenna pattern frame can be obtained by equation (10). The observations of the Moon projected in antenna pattern frame are plotted in Figure 4 (a), (c), (e). These observations are fitted by using 2D Gaussian function (Equation 14) and the deviation of the center ($x_0, y_0$) are used to calculate the cost function.

In brief, the algorithm doesn't require that the Moon and the satellite stay still during the lunar scan. From the Moon ephemeris, the GPS observations, and a series of rotation matrices in geolocation process, the position of the Moon where it appears in the antenna pattern frame at any observation time can be accurately determined.

As the way to calculate the Moon vector in ECI through NOVAS package has already been implemented in ADL, to keep the paper concise, the detailed description of the process is not added to the manuscript. Instead, the application of the package in ADL will be mentioned for reference. The following statements are added to Section 3.1:

"Given the ephemeris of the Moon and the satellite position vector provided by GPS, the lunar vector ($l_{ECI}$) originating from satellite to the Moon in ECI at any observation time can be calculated through the Naval Observatory Vector Astrometry Software Package version F3.1 (NOVAS F3.1). This software package is an open source library for computing various commonly used quantities in positional astronomy (Kaplan et al., 2011). It has been implemented in the Algorithm Development Library (ADL), a software program for processing the NOAA polar-orbiting satellite data, to calculate the Sun and Moon vectors in order to determine the zenith and azimuthal angles of the Sun and set the lunar intrusion flag (https://jpss.ssec.wisc.edu/)."

2. The authors give no explanation for the surprising fact that the boresight pointing error of channels 1 - 15 in pitch direction is three times as high as all other errors. Could it be that this error is an artefact caused by the time constant of the receiver V and K/Ka? If the devices react with a little delay to changes in the incoming flux, then they will give their maximum signal only after the Moon was at the origin of the antenna pattern frame.

Reply: Thank you for your comment. We've noticed this issue and pointed it out in the manuscript. The receiver response delay seems not like the major cause. ATMS is a cross-track scanning radiometer. The response delay of its receivers will not only lead to the maximum signal deviation in along-track direction but also in cross-track direction.

The aim of the lunar algorithm developed in this manuscript is to estimate the total static geolocation error contributed by all the possible static error sources, such as antenna beam misalignment and instrument mounting error caused by the on-orbit thermal dynamic change and shift during the launch. As for quantitatively identifying the bias caused by each internal interface alignment, though it is very helpful for finding out the instrument hardware deficiency, it is beyond the capability of this algorithm and beyond the scope of this study. To achieve that goal, other new methods need to be developed and we will work on that topic in our future study.

The following discussion has been added to the end of the first paragraph in section 5:

"As the coastline inflection point method, the lunar scan method estimates the boresight pointing error caused by all the possible instrument interfaces, such as the antenna beam misalignment and the instrument mounting error. To further identify the amount of bias each internal interface of the instrument could cause is beyond the capability of the lunar scan algorithm."

Technical Corrections:

Page 1, line 15: The authors only explain what the "S" in "SNPP" stands for - it is better to explain the whole abbreviation.

The full name for SNPP, "Suomi National Polar-orbiting Partnership" is added. Thank you for pointing out this.

Page 2, lines 17 and 19: I suggest to cite also R. Bonsignori, In-orbit verification of microwave humidity sounder spectral channels coregistration using the moon, Applied Remote Sensing, Volume 12, id. 025013 (2018).

Reply: Thank you very much for your suggestion. This article is very innovative and inspiring. We've cited it and added some more discussion about the application of Lunar intrusion to geolocation validation based on it. The 4th paragraph of Section 1 is rewritten as follows:

"However, the possibility of using the Moon for geolocation validation and correction of microwave sensors has not been widely discussed. Attempt has been made to utilize the lunar intrusion data to assess the boresight pointing error of Microwave Humidity Sounder (MHS) (Burgdorf et al., 2016,) and ATMS (Zhou et al., 2017), and the mutual alignment of channels of MHS (Bonsignori et al., 2018). The advantage of using lunar intrusion data to assess the beam misalignment is that it won't interrupt the routine operations of the satellite. However, the disadvantage is only the along-track component of the misalignment can be obtained, since the Moon only passes four space view pixels sampled 1.1° apart along the cross-track direction (Bonsignori et al., 2018). Besides, for ATMS, it has already been proved that its boresight pointing error is scan-angle dependent (Zhou et al., 2019). The boresight deviation estimated at space views thus cannot represent that at the Earth views which is truly valuable for the on-orbit geolocation accuracy enhancement."

Page 2, lines 19 and 20: I do not understand "the Moon only passes the along-track direction".

Reply: This statement has been revised to:

"However, the disadvantage is that only the along-track component of the misalignment can be obtained, since the Moon only passes four space view pixels sampled 1.1° apart along the cross-track direction (Bonsignori et al., 2018)."

Thank you!

Page 2, line 31: There is no Sect. 6.

Sorry for this typo. It is corrected as follows:

"This paper is organized as follows: Sect. 2 briefs ATMS instrument scan geometry and geolocation process, followed by presenting the lunar scan observations captured by ATMS during the pitch-over maneuver operation, Sect.3 describes the methodology, Sect. 4 presents the results and the validation with the coastline method developed in (Zhou et al., 2019), conclusions and discussions are in Sect. 5"

Page 7, line 16: What are "negative observations"?

Reply: The negative observations are caused by observation noise. As described in the last paragraph of Section 2, after calibration, correction for warm bias, earth side lobe contamination and reflector emission contamination, and removal of the cosmic background radiation, the observations only contain the lunar signal and random noise. For the pixel where the Moon is far away from beam center, the signal is so weak that its magnitude is exceeded by that of the noise. Under this circumstance, the observations may have negative values as showed in Figure 2 of the manuscript and these observations should be removed from the data set.

Figure 4: Units of axes?

Reply: The axes of Figure 4 are unitless. The labels of the axes are updated:

[Figure]

Page 9, line 15: "lower" instead of "higher"?

Yes. We have corrected it. Thank you!

Figure 5: The red X is hard to see.

The red X has been made bold. Thanks!

[Figure]

Figure 7: The colours of the two Xs are almost the same.

The color of the X for pitch is changed to green to enhance the contrast. Thanks!

[revised manuscript text omitted]

---

## Author Comment (AC3) · 18 Jun 2019

Dear Referee 2 and Editors,

Thank you very much for your valuable comments and suggestions. The reply to them along with the revised manuscript have been combined to a single pdf and uploaded as a Supplement. Please do not hesitate to contact me (jzhou128@umd.edu) if you have any questions.

Best regards,

Jun Zhou and Hu Yang 6/18/2019

[Figure]

Please also note the supplement to this comment:
https://www.atmos-meas-tech-discuss.net/amt-2019-177/amt-2019-177-AC3-
supplement.pdf

―――――――――――――――――――

**Supplement:**

General Comments

This paper gives a good overview of the application of using the moon as a calibration reference for determining geolocation, which is necessary for microwave radiometers and as noted by the authors is difficult for channels that do not see Earth's surface. A detailed explanation of the method is given and results shown applying the method to ATMS. In addition to the comments already noted by the other reviewers, I have the following comments below.

Thank you very much for your valuable comments and suggestions. They are very helpful for us to improve the lunar scan algorithm and the quality of the manuscript. We've studied your comments thoroughly and been able to respond to each of them as below. The manuscript has been revised accordingly and the changes made to it have been marked by using latexdiff.

Specific Comments

The Figure 7 coastline Euler angles do not match those shown in Zhou et al. 2019 Figure 9. Can you please provide an explanation for this? Since Figure 7 provides validation for your lunar method and makes a direct reference to the Zhou et al. 2019 paper, I'm assuming Figure 9 in that paper is what you are referring to and these figures should then match unless you explain the discrepancies.

Reply: Yes, the Euler angles of ATMS presented in Figure 7 of this manuscript are different from those in Figure 9 of Zhou et al. 2019. This is because they are the retrieval results for ATMS onboard different satellites. This manuscript focuses on NOAA-20, whereas Zhou et al. 2019 focuses on SNPP. Though for the same instruments, due to the different installation process, on-orbit thermal dynamic change and shift during the launch the two satellites have gone through, the boresight pointing errors of them could be different.

Another comment in regards to Figure 7, is that it would be good to show a table listing the coastline Euler angles from your previous paper at FOV 66, compared with these numbers along with the difference to quantitatively show how the methods compare rather than just the figure.

Reply: Thank you for your suggestion. The retrieval results of the two methods and the absolute difference between them are presented in Table 1. This table is added to the revised manuscript right after Figure 7.

| Ch. | Lunar Scan Method | | Coastline Method | | Difference | |
|-----|---------|----------|----------|----------|----------|----------|
| | Roll (°) | Pitch (°) | Roll (°) | Pitch (°) | Roll (°) | Pitch (°) |
| 1 | 0.05 | 0.22 | -0.01 | 0.25 | 0.06 | 0.03 |
| 2 | -0.07 | 0.25 | -0.02 | 0.34 | 0.05 | 0.09 |
| 3 | 0.0 | 0.25 | 0.01 | 0.28 | 0.01 | 0.03 |
| 16 | -0.07 | -0.08 | -0.03 | -0.03 | 0.04 | 0.05 |
| 17 | -0.04 | 0.02 | None | None | None | None |

Table 1 Euler angles at FOV 66 retrieved from lunar scan method and coastline inflection point method as well as the absolute difference between them.

The following discussion is also added to the end of the third paragraph of Section 4:

"The numbers are also listed in Table 1.

The correlation between the Euler angles retrieved from these two independent methods is evident. Considering the RMSE of the fitting in coastline method is 0.05 ° on average, the difference between the two methods are close to the uncertainty of the coastline method except for the pitch of channel 2. The lunar scan algorithm can measure the on-orbit instrument boresight pointing error to subpixel accuracy. For the sounding channels where the coastline method is inapplicable, the lunar scan algorithm is still capable of delivering reasonable estimations of the geometric calibration errors. The retrieval error of the lunar scan method could be mainly caused by the instrument noise and the irregularities of the antenna pattern. With more lunar scan data accumulated in the future, the uncertainty of the method can be better estimated and reduced."

The abstract and conclusion section are revised accordingly.

Technical Corrections

Page 2, line 7. Change "matric" to "metric".

What we are talking about here is the correction matrices that could be applied to operational geolocation process to enhance the on-orbit geolocation accuracy. The "matric" is corrected to "matrices". Thanks!

Page 4, line 3. Change "showed" to "shown".

Corrected. Thanks!

Page 6, line 2. Change "locating" to "located".

Corrected. Thanks!

Page 6, line 11. Change "showed" to "shown".

Corrected. Thanks!

Page 8, line 6. Change "locating" to "located".

Corrected. Thanks!

Page 9, line 22. Change "corresponding" to "correspond".

Corrected. Thanks!

[revised manuscript text omitted]